# A single clonal lineage of transmissible cancer identified in two marine mussel species in South America and Europe

Marisa A Yonemitsu[1], Rachael M Giersch[1], Maria Polo-Prieto[2], Maurine Hammel[3,4], Alexis Simon[3], Florencia Cremonte[5], Fernando T Avilés[6], Nicolás Merino-Véliz[6], Erika AV Burioli[7], Annette F Muttray[8], James Sherry[9], Carol Reinisch[9], Susan A Baldwin[10], Stephen P Goff[2,11,12], Maryline Houssin[7,13], Gloria Arriagada[6], Nuria Vázquez[5], Nicolas Bierne[3], Michael J Metzger[1]*

[1]Pacific Northwest Research Institute, Seattle, United States; [2]Howard Hughes Medical Institute, Chevy Chase, United States; [3]ISEM, Université de Montpellier, CNRS- EPHE-IRD, Montpellier, France; [4]IHPE, Université de Montpellier, CNRS-Ifremer-UPVD, Montpellier, France; [5]Laboratorio de Parasitología (LAPA), Instituto de Biología de Organismos Marinos (IBIOMAR) (CCT CONICET - CENPAT), Puerto Madryn, Argentina; [6]Instituto de Ciencias Biomedicas, Facultad de Medicina y Facultad de Ciencias de la Vida, Universidad Andres Bello, Santiago, Chile; [7]Research and Development, LABÉO Frank Duncombe, Saint-Contest, France; [8]Environmental Resources Management, Vancouver, Canada; [9]Water Science & Technology Directorate, Environment and Climate Change Canada, Burlington, Canada; [10]Chemical and Biological Engineering, University of British Columbia, Vancouver, Canada; [11]Department of Microbiology and Immunology, Columbia University Medical Center, New York, United States; [12]Department of Biochemistry and Molecular Biophysics, Columbia University Medical Center, New York, United States; [13]FRE BOREA, MNHN, UPMC, UCN, CNRS-7208, IRD-207, Université de Caen Normandie, Caen, France

*For correspondence:
metzgerm@pnri.org

Competing interests: The authors declare that no competing interests exist.

**Abstract** Transmissible cancers, in which cancer cells themselves act as an infectious agent, have been identified in Tasmanian devils, dogs, and four bivalves. We investigated a disseminated neoplasia affecting geographically distant populations of two species of mussels (*Mytilus chilensis* in South America and *M. edulis* in Europe). Sequencing alleles from four loci (two nuclear and two mitochondrial) provided evidence of transmissible cancer in both species. Phylogenetic analysis of cancer-associated alleles and analysis of diagnostic SNPs showed that cancers in both species likely arose in a third species of mussel (*M. trossulus*), but these cancer cells are independent from the previously identified transmissible cancer in *M. trossulus* from Canada. Unexpectedly, cancers from *M. chilensis* and *M. edulis* are nearly identical, showing that the same cancer lineage affects both. Thus, a single transmissible cancer lineage has crossed into two new host species and has been transferred across the Atlantic and Pacific Oceans and between the Northern and Southern hemispheres.
DOI: https://doi.org/10.7554/eLife.47788.001

## Introduction

Cancers normally arise from mutations in an organism's own cells, and they either regress or continue to grow until they kill the host organism. In a few cases, however, including Tasmanian devils

**eLife digest** Cancer cells can grow and spread in one individual, but they normally do not spread to others. There are a few exceptions to this rule. For example, there are cancers in Tasmanian devils, dogs and bivalve shellfish that can spread to other members of the same species. In these creatures, cancer from one individual evolved the ability to spread throughout the population. These cancer cells infect animals like a pathogen.

A fatal cancer called disseminated neoplasia affects many species of bivalves. In four bivalve species, including the marine mussel *Mytilus trossulus*, scientists have shown that the cancer can spread from one individual to another. This transmissible cancer has been found in *M. trossulus* mussels in British Columbia, Canada; but related species of mussels in other parts of the world also develop disseminated neoplasia. It is possible these other cancers are transmissible and have spread from one population of mussels to another.

Yonemitsu et al. performed genetic analyses to show that cancers found in two other mussel species – *Mytilus chilensis* in South America and *Mytilus edulis* in Europe – are transmissible and arose in *M. trossulus.* The cancers in the South American and European mussels were nearly identical genetically, which suggests that they came from a single *M. trossulus* mussel with cancer at some point in the past. Somehow cancer cells spread between the Northern and the Southern Hemispheres and across the Atlantic Ocean, infecting multiple species across the world. The analyses also show that this cancer lineage is different from the one previously identified in British Columbia.

These analyses show that bivalve transmissible neoplasia was able to spread worldwide, most likely through accidental transport of infected mussels on international shipping vessels. This suggests that human activities unwittingly introduced the disease to new areas. Learning more about transmissible cancers may help scientists understand how cancers evolve with their hosts in extreme situations.

DOI: https://doi.org/10.7554/eLife.47788.002

(*Pearse and Swift, 2006*; *Pye et al., 2016*), dogs (*Murgia et al., 2006*; *Rebbeck et al., 2009*), and bivalves (*Metzger et al., 2016*; *Metzger et al., 2015*), cancer cells naturally transfer from one animal to another as an infectious allograft, leading to lineages of transmissible cancer that spread through the populations (*Metzger and Goff, 2016*). In bivalves, disseminated neoplasia (also called hemic neoplasia) is a leukemia-like disease that has been found in more than a dozen marine species, characterized by an amplification of rounded, polyploid cells in the hemolymph of animals that infiltrate into tissues. This disease often occurs in outbreaks and has led to significant population losses in many bivalve species (*Barber, 2004*; *Carballal et al., 2015*). Pollution and retroviruses had been investigated as suspected causes, but until recently the etiology of disseminated neoplasia in bivalves was unknown. In four of the bivalve species affected by disseminated neoplasia (soft-shell clams, *Mya arenaria*; cockles, *Cerastoderma edule*; golden carpet shell clams, *Polititapes aureus*; and bay mussels, *Mytilus trossulus*), recent investigation has shown that the diseases are bivalve transmissible neoplasias (BTNs), showing that transmissible cancer may, in fact, be a common phenomenon in marine invertebrates.

Disseminated neoplasia has been reported in marine mussels as early as 1969 (*Farley, 1969*), and has been observed consistently throughout Europe in both *M. edulis* (*Green and Alderman, 1983*; *Lowe and Moore, 1978*; *Rasmussen, 1986*) and *M. galloprovincialis* (*Ciocan and Sunila, 2005*; *Carella et al., 2013*; *Gombač et al., 2013*), albeit at lower prevalence than has been observed in some *M. trossulus* populations, which reached >20% in some locations in the 1980s and was associated with population losses (*Barber, 2004*; *Bower et al., 1994*). Mussels in the genus *Mytilus* form a complex of species, including *M. trossulus*, *M. edulis*, and *M. galloprovincialis*, native to the northern hemisphere, and *M. chilensis, M. platensis, and M. planulatus*, native to the southern hemisphere (*Zbawicka et al., 2018*). These species have an anti-tropical distribution, spanning the temperate waters of the world, and they can hybridize, leading to widespread introgression of allospecific alleles (*Fraïsse et al., 2016*) as well as hybrid zones in several locations (*Bierne et al., 2003*). Disseminated neoplasia has been reported in four of these *Mytilus* species across multiple continents

(*Barber, 2004*; *Carballal et al., 2015*), but so far the only lineage of disseminated neoplasia in the genus that has been directly confirmed to be a transmissible cancer is in *M. trossulus* along the Pacific coast of British Columbia (BC), Canada (*Metzger et al., 2016*).

A population genetics analysis of *M. edulis* from France, conducted to detect hybridization and introgression between *Mytilus* species by detecting species-specific SNPs, identified a few individuals with unexpected 'chimeric' signals. The DNA samples from these individuals included *M. trossulus* alleles in a population where that species was absent, and the ratio of the *M. edulis* and *M. trossulus* alleles were not consistent with a diploid hybrid genome. Instead, this could be better explained by a mixture of two genotypes (one *M. edulis* and one *M. trossulus* in origin) within a single individual (*Riquet et al., 2017*). This suggested the hypothesis that *M. edulis* in France are affected by a BTN originating from *M. trossulus*, possibly the same lineage as previously identified in Canada. More recently, analysis of a severe mass mortality event (*Benabdelmouna and Ledu, 2016*) in French mussels that began in 2014 led to the observation of disseminated neoplasia in many populations of *M. edulis* and *M. galloprovincialis* (*Benadelmouna et al., 2018*), although it has not yet been determined whether this neoplasia is transmissible.

*M. chilensis* mussels are found along the western coast of South America, and disseminated neoplasia has been reported in multiple populations in Chile and Argentina since 1998 (*Lohrmann et al., 2019*; *Cremonte et al., 2015*; *Campalans et al., 1998*; *Cremonte et al., 2011*). Prevalence has varied by location, ranging from undetectable in some populations to as high as 12% in the Castro region of Chile and 13% in the Beagle Channel in southern Argentina, although no notable mass mortality events have been associated with neoplasia in these populations.

We investigated disseminated neoplasia found in *M. chilensis* from Argentina and Chile, as well as *M. edulis* from France and the Netherlands, to determine if these cancers are transmissible cancers or conventional cancers and, if transmissible, whether these cancers represent new cases of the same lineage of transmissible cancer previously reported in Canada (here termed *Mytilus* BTN1) or whether they are novel cancer lineages. *Mytilus* BTN1 was able to be distinguished from normal *M. trossulus* genotypes at both a nuclear and mitochondrial locus (*Metzger et al., 2016*). Through analysis of multiple genetic markers in South American *M. chilensis* and European *M. edulis* individuals, we found evidence of transmissible cancer in both populations. We found that neither cancer matched the *Mytilus* BTN1 lineage, previously found in BC *M. trossulus*, but instead the cancers in both species appear to come from a single, previously unidentified transmissible cancer, here called *Mytilus* BTN2 or *Mtr*BTN2. This BTN lineage arose from an *M. trossulus* individual, and it has since crossed into two different *Mytilus* species and is now affecting animals across both the Atlantic and Pacific Oceans and in both the Northern and Southern Hemispheres.

## Results

### Transmissible neoplasia in *M. chilensis*

To determine whether the disseminated neoplasia in *M. chilensis* was due to a conventional cancer or a transmissible cancer lineage, we collected 60 *M. chilensis* animals from Argentina in 2012. Histological analysis was used to diagnose the animals for the presence of disseminated neoplasia, and we observed a disease prevalence of 10%. DNA was extracted from tissues of three diseased animals and three normal animals, and an intron-spanning region in the nuclear gene Elongation Factor one alpha (*EF1α*) was amplified and sequenced. A normal individual would be expected to have one or two alleles (depending on whether it was homozygous or heterozygous at that locus), and a cancer would be expected to have its own set of alleles, derived from its original host. All three normal animals had one or two alleles, as expected, but two of the diseased animals (Mch41 and Mch42) had four distinct alleles, consistent with a mixture of two host alleles and two cancer-specific alleles (*Figure 1A*). Notably, each diseased sample had two alleles that were unique (corresponding to host alleles), but two alleles from animal Mch41 were nearly exact matches to two alleles from Mch42 (here notated as G and H). This is consistent with the presence of a transmissible cancer containing alleles G and H. These cancer-associated alleles were not cloned from one of the three diseased animals (Mch23), but cloning of PCR products is not very sensitive. This individual either does not have this cancer lineage or has a lower level of cancer that was not detectable with this method.

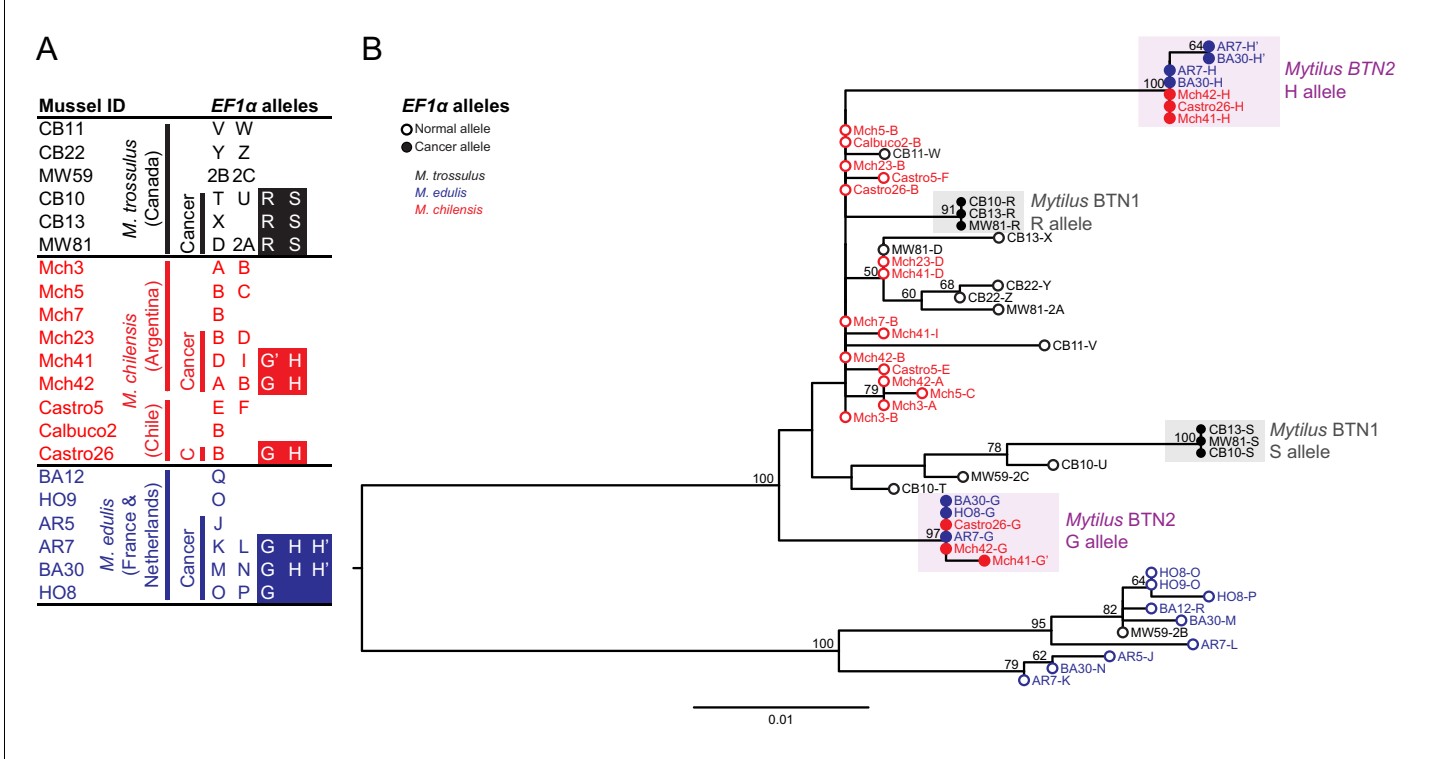

**Figure 1.** Phylogenetic analysis of *EF1α* alleles from normal and diseased mussels. The *EF1α* locus was amplified, and multiple alleles were cloned from different individual normal and diseased mussels of different species and locations: *M. trossulus* from BC (black), *M. chilensis* from Argentina and Chile (red), and *M. edulis* from France and the Netherlands (blue). (**A**) A list of cloned alleles is shown, with filled boxes marking cancer-associated alleles. (**B**) Phylogenetic analysis of aligned alleles shows groups of related alleles (see *Figure 1—source data 1*). Names of alleles on the tree specify individual ID and allele ID. Open circles mark alleles from normal individuals and host alleles from diseased individuals. Filled circles mark cancer-associated alleles on the tree (colored by host species). The tree was rooted at the midpoint, with bootstrap values below 50 removed. Model used was HKY85+G. The scale bar marks genetic distance.

DOI: https://doi.org/10.7554/eLife.47788.003

The following source data is available for figure 1:

**Source data 1.** FASTA formatted text file of *EF1α* sequence alignment.

DOI: https://doi.org/10.7554/eLife.47788.004

A second collection of farmed *M. chilensis* from two locations in Chile confirmed the presence of disseminated neoplasia in these populations (9.6% in Calbuco and 4.5% in Castro). *EF1α* alleles cloned from a diseased individual from this location exactly match the G and H cancer-associated sequences from Argentina.

Phylogenetic analysis of these putative cancer-associated alleles shows that they are likely of *M. trossulus* or *M. chilensis* origin (*Figure 1B*). However, both cancer-associated alleles are clearly distinct from normal *M. trossulus* individuals and distinct from either allele of the previously identified BC transmissible neoplasia, *Mytilus* BTN1. Additionally, phylogenetic analysis shows that the *EF1α* S allele from *Mytilus* BTN1 (previously termed the 'minor' allele) is closely related to a subset of normal *M. trossulus* alleles. Thus, the tree is incompatible with the cancer-associated G and H alleles arising from the same cancer lineage as the *Mytilus* BTN1 S allele, although analysis of additional loci is needed to confirm this finding. The BTN found in *M. chilensis* therefore likely represents a separate, independent lineage of transmissible cancer, here called *Mytilus* BTN2.

## Common transmissible cancer lineage in *M. chilensis* and *M. edulis*

The first evidence that disseminated neoplasia in European *M. edulis* was a BTN was the observation of 'chimeric' signals in a SNP detection assay, in which SNPs specific for *M. trossulus* were detected in five *M. edulis* samples out of 938. Additionally, the *M. trossulus* SNPs were detected at

intermediate fluorescence ratio that could not be explained by an individual that was homozygous or heterozygous at those loci (i.e. variant allele fractions were not close to 50% or 100%) (*Riquet et al., 2017*). We sequenced the *EF1α* alleles present in DNA from four of these 'chimeric' samples of *M. edulis* collected from locations on the Atlantic coast of Europe and found more than the normal two alleles in three of them (*Figure 1A*). Only one or two alleles were observed in normal *M. edulis* from the same populations. Two alleles were shared in more than one diseased sample, suggesting that they could be from a transmissible cancer. Unexpectedly, these two cancer-associated alleles found in diseased *M. edulis* did not match the previously reported *Mytilus* BTN1 lineage, found in *M. trossulus* in Canada, but instead exactly matched those found in cancer samples from *M. chilensis* in South America (*Mytilus* BTN2) (*Figure 1B*).

In these European *M. edulis* cancer samples, we found alleles identical to the G allele from *M. chilensis* and two nearly identical versions of the H allele, with only one SNP between them (termed H'). Both the H and H' alleles were represented by multiple clones and found in multiple individuals, so H' likely represents a true second copy of the H allele within the cancer cells. Disseminated neoplasia is polyploid, and this result is consistent with a de novo mutation which occurred in one copy of the H allele after genome duplication (and likely after divergence of the South American and European strains of this lineage).

## Confirmation of cancer lineage identity with nuclear and mitochondrial loci

To gain more information about identity of the cancer lineage, we sequenced three additional loci using primers which amplify alleles in all three *Mytilus* species tested here. Histone 4 (*H4*) is a conserved gene present in multiple copies in the nuclear genome (*Eirín-López et al., 2004*). We amplified and cloned alleles from *M. chilensis* and *M. edulis* samples, as done with *EF1α*. We observed 2–5 alleles in diseased samples and only 1–2 in normal samples (*Figure 2A*). Looking only at normal alleles, there is a clear grouping based on species of origin in a phylogenetic tree (*Figure 2B*). There are three exceptions to this clustering, from one normal *M. trossulus* (MW59) and two diseased *M. chilensis* (Mch41 and Castro26) with some distinctly *M. edulis*-like alleles. These may reflect introgression or true hybrid individuals—MW59 also has an *EF1α* allele corresponding to *M. edulis* and likely represents a hybrid with this species, which has been introduced into Canada (*Crego-Prieto et al., 2015*).

Two sets of cancer-associated alleles were identified from both *M. chilensis* and *M. edulis*, which group together. As with *EF1α*, the alleles associated with cancer in *M. chilensis* and *M. edulis* are distinct from the sequences of *Mytilus* BTN1 found in *M. trossulus* from Canada. Overall, the *H4* locus has less variability than the *EF1α* locus. Notably, there are alleles found in normal *M. trossulus* individuals that are exact matches to alleles found in both *Mytilus* BTN1 and *Mytilus* BTN2. These data again argue that this newly identified cancer lineage has an independent *M. trossulus* origin and that it arose from a different *M. trossulus* individual than the one that gave rise to *Mytilus* BTN1. *Mytilus* BTN1 likely arose within an *M. trossulus* individual with a 2A *H4* allele and Mytilus BTN2 arose from a different *M. trossulus* individual which carried both MR-like and KNS-like alleles.

In addition to the nuclear loci, we also amplified a control region of the mitochondrial genome (mtCR) including large subunit ribosomal RNA (lrRNA), a variable domain, and conserved domain (*Burzyński et al., 2006*). As *Mytilus* mussels have double uniparental inheritance of mitochondria (in which all individuals inherit mitochondria maternally, but a second male-specific lineage of mitochondria is also passed down to male offspring gonads), we amplified a female allele from all individuals, and both female and male alleles from about half of the individuals (10 of 21 total) (*Figure 3A*). The sequences of normal *M. trossulus* and *M. edulis* are similar to previously published reference sequences from those species. In addition, we amplified a *M. trossulus*-like allele from cancer samples from both *M. edulis* and *M. chilensis*. This allele (termed 'D') is mostly female-like, but it switches to male-like sequence at the 3' end (*Figure 3B*). This recombinant sequence closely matches a recombinant sequence previously reported in a Baltic *M. trossulus* individual (*Śmietanka and Burzyński, 2017*; *Zbawicka et al., 2014*). Interestingly, a second *M. trossulus*-like allele (C) was found in all three cancer samples of *M. chilensis*, which also represents a recombination between female and male sequences, but with a different recombination point than the first. This argues for heteroplasmy of the cancer cells in *M. chilensis*, and either loss of this heteroplasmy in the cancer lineage infecting *M. edulis* or capture of this new mitochondrial genome in the cancer lineage infecting *M. chilensis*.

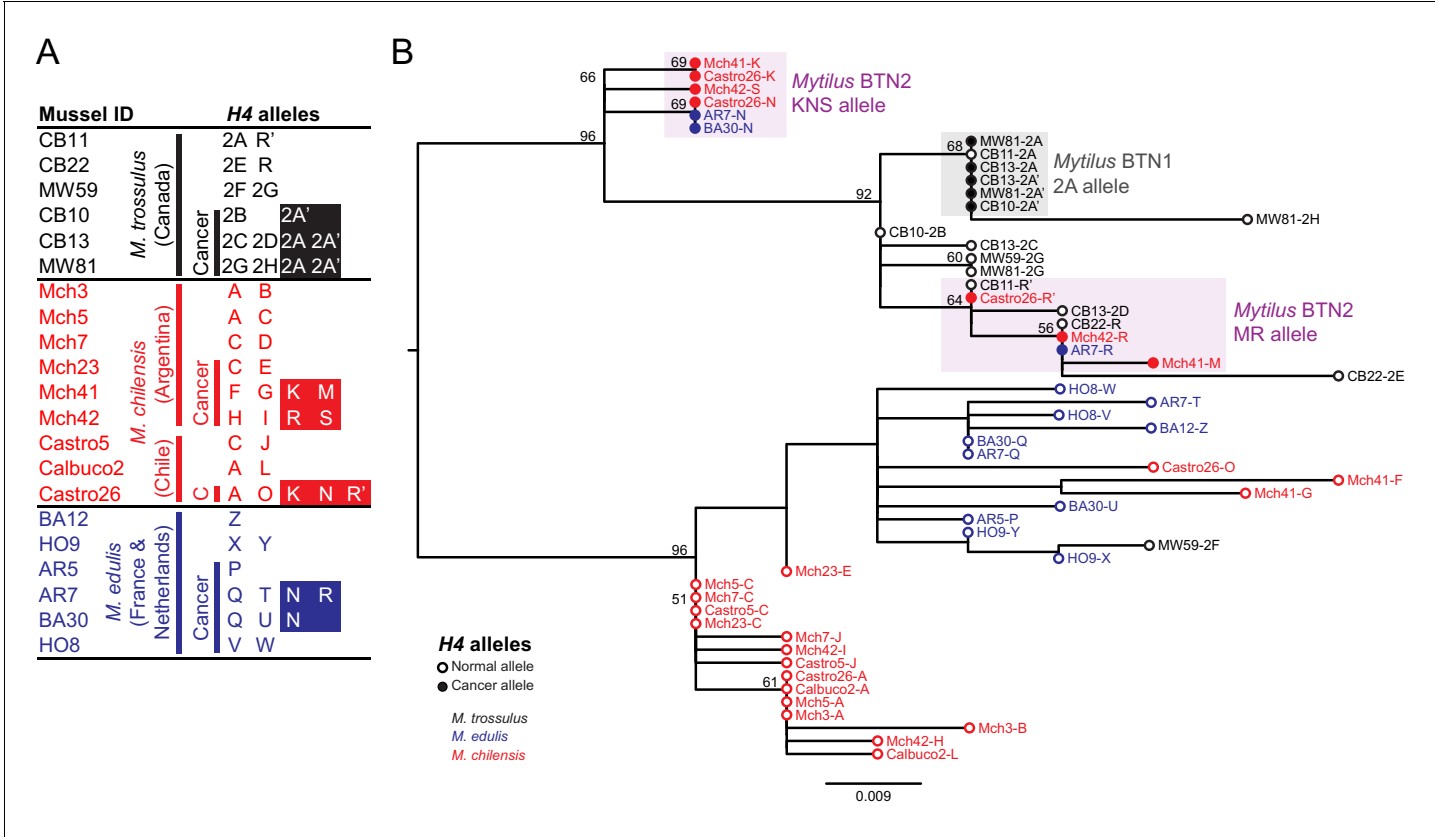

**Figure 2.** Phylogenetic analysis of *H4* alleles from normal and diseased mussels. The *H4* locus was amplified, and multiple alleles were cloned from different individual normal and diseased mussels of different species and locations: *M. trossulus* from BC (black), *M. chilensis* from Argentina and Chile (red), and *M. edulis* from France and the Netherlands (blue). (**A**) A list of cloned alleles is shown, with filled boxes marking cancer-associated alleles. (**B**) Phylogenetic analysis of aligned alleles shows groups of related alleles (see *Figure 2—source data 1*). Names of alleles on the tree specify individual ID and allele ID. Open circles mark alleles from normal individuals and host alleles from diseased individuals. Closed circles mark cancer-associated alleles (colored by host species). The tree was rooted at the midpoint, with bootstrap values below 50 removed. Model used was HKY85+I. The scale bar marks genetic distance. Three sequences (AR7-2I, BA30-2J, and Castro26-2J) are consistent with recombination between the two cancer-associated alleles and were removed from the tree for clarity. The two alleles identified in the *Mytilus* BTN1 lineage (2A and 2A') are distinguished by a unique single base deletion in 2A', which does not show up as a difference in the tree, as gaps are treated as missing data.

DOI: https://doi.org/10.7554/eLife.47788.005

The following source data is available for figure 2:

**Source data 1.** FASTA formatted text file of *H4* sequence alignment.

DOI: https://doi.org/10.7554/eLife.47788.006

These unique recombinants between female and male sequences again argue that *Mytilus* BTN1 and BTN2 represent distinct lineages of cancer. Even when considering only the region without any recombination with the male sequence (*Figure 3C*), both of these mitochondrial alleles are distinct from those found in normal animals and from the previously identified *Mytilus* BTN1 lineage, and the phylogenetic tree suggests that *Mytilus* BTN1 and BTN2 arose from different *M. trossulus* individuals.

We additionally sequenced a second region in the mitochondrial DNA, the cytochrome c oxidase I gene (*COI*). Standard 'universal' molluscan primers used for barcoding (*Folmer et al., 1994*) did not amplify the mt*COI* allele from *Mytilus* BTN2, but by using degenerate primers flanking the region, we were able to amplify and sequence mt*COI* from all samples (*Figure 4A*, *Supplementary file 1*). The mt*COI* of the male-specific mitogenome was not amplified by these primers. We identified a clear separation of the mitogenomes of all three species, with the mt*COI* from both *Mytilus* BTN1 and 2 nested within the *M. trossulus* sequences, as with the three other loci tested (*Figure 4B*). The *Mytilus* BTN1 allele (U) again showed clear differences from the *Mytilus*

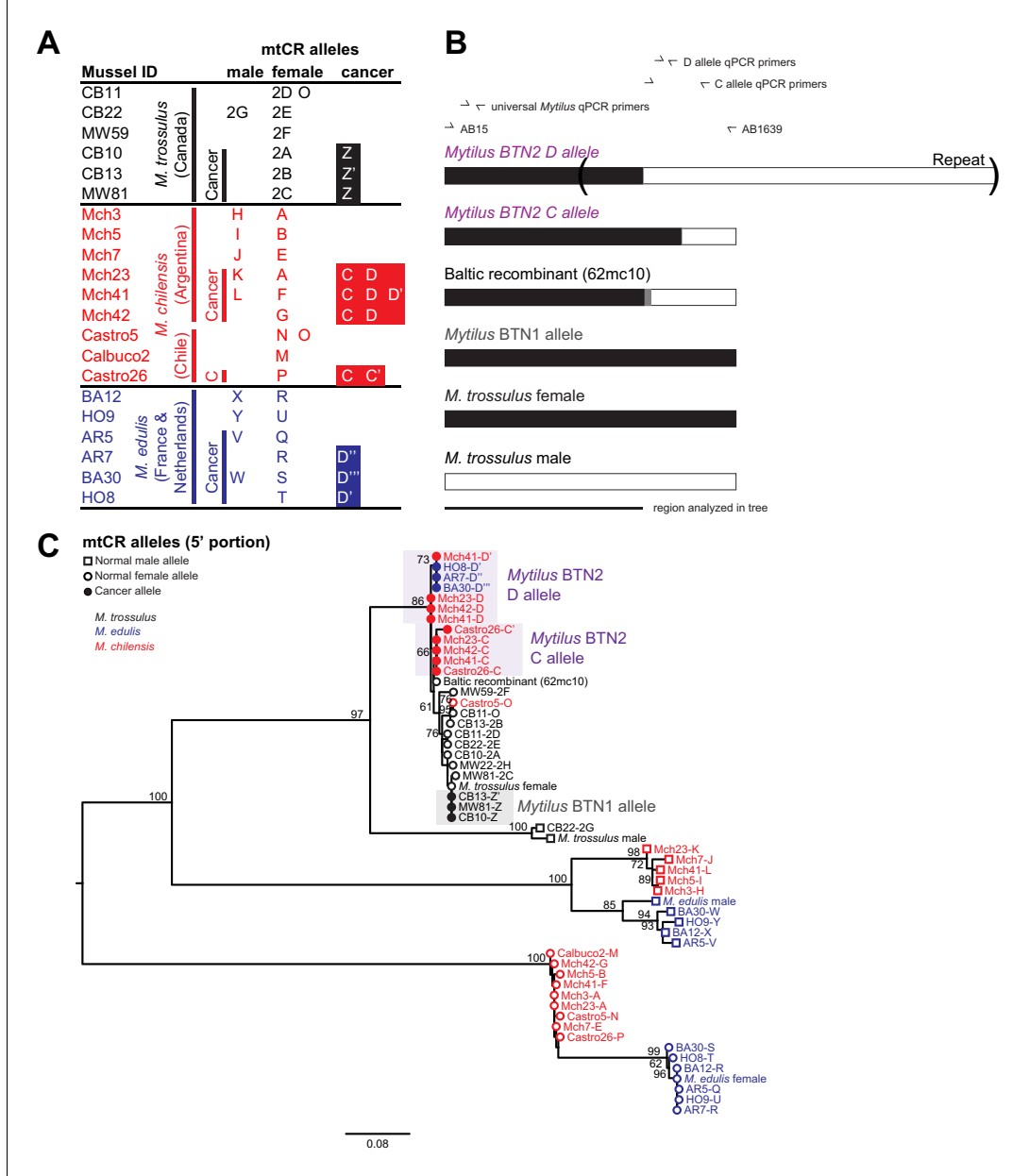

**Figure 3.** Phylogenetic analysis of mitochondrial CR alleles from normal and diseased mussels. The CR region of mitochondrial DNA was amplified, and multiple alleles were cloned from different individual normal and diseased mussels of different species and locations: *M. trossulus* from BC (black), *M. chilensis* from Argentina and Chile (red), and *M. edulis* from France and the Netherlands (blue). (**A**) A list of cloned alleles is shown, with filled boxes marking cancer-associated alleles. (**B**) A schematic of the rearrangement found in the cancer-associated alleles is shown with black boxes representing female-derived sequence and white boxes representing male-derived sequence (gray bars represent the likely recombination region). Parentheses mark the repeat region in allele D. (**C**) Phylogenetic analysis of aligned alleles shows groups of related alleles (see *Figure 3—source data 1*). Names specify individual ID and allele ID. Alleles from normal individuals and host alleles from diseased individuals are marked with open circles (female allele) and open squares (male allele). Closed circles mark cancer-associated alleles (colored by host species). The tree was rooted at the midpoint, with bootstrap values below 50 removed. Model used was GTR+I. The scale bar marks genetic distance. Multiple male alleles were identified in *M. chilensis* samples, and a representative with the greatest number of clones was chosen for each individual. Reference sequences include female *M. trossulus* (AY823625.1), male *M. trossulus* (HM462081.1), female *M. edulis* (DQ198231.2), male *M. edulis* (AY823623.1), and the *M. trossulus* recombinant 62cm10 (KM192133.1).
DOI: https://doi.org/10.7554/eLife.47788.007

The following source data is available for figure 3:

**Source data 1.** FASTA formatted text file of mtCR sequence alignment.
DOI: https://doi.org/10.7554/eLife.47788.008

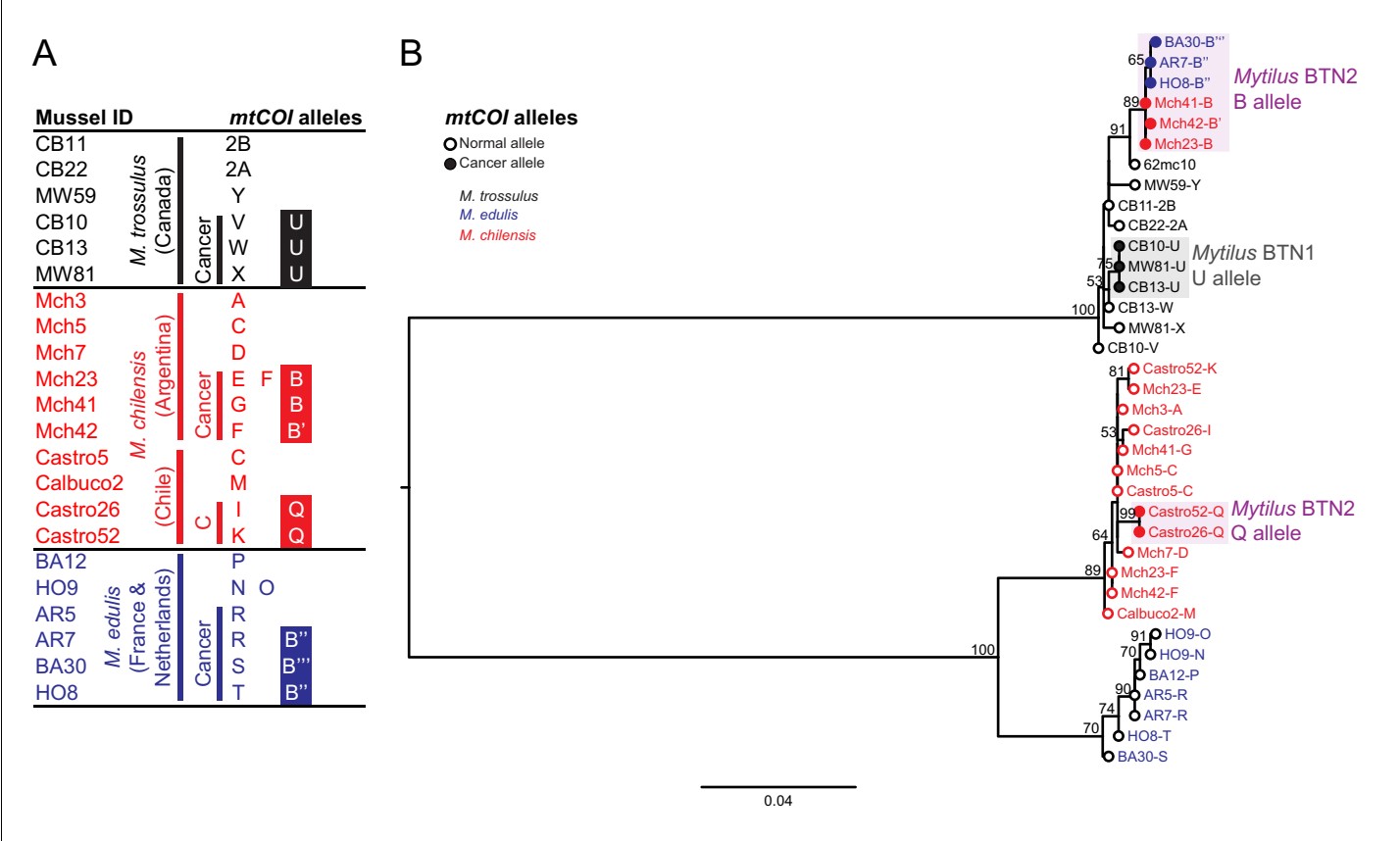

**Figure 4.** Phylogenetic analysis of mitochondrial *COI* alleles from normal and diseased mussels. The mt*COI* locus was amplified, and multiple alleles were cloned from different individual normal and diseased mussels of different species and locations: *M. trossulus* from BC (black), *M. chilensis* from Argentina and Chile (red), and *M. edulis* from France and the Netherlands (blue). (**A**) A list of cloned alleles is shown, with filled boxes marking cancer-associated alleles. (**B**) Phylogenetic analysis of aligned alleles shows groups of related alleles (see *Figure 4—source data 1*). Names of alleles on the tree specify individual ID and allele ID. Open circles mark alleles from normal individuals and host alleles from diseased individuals. Closed circles mark cancer-associated alleles (colored by host species). The tree was rooted at the midpoint, with bootstrap values below 50 removed. Model used was GTR+G. The scale bar marks genetic distance. An additional Chilean sample was used for cloning (Castro52) as Castro26 showed difference from the other samples at this locus and a second sample was needed to infer the cancer-associated allele and confirm the mitochondrial replacement. One clone obtained from Castro26 was a recombinant between a *M. trossulus*-like sequence and an *M. chilensis*-like sequence, and it was excluded from the phylogenetic tree for clarity.

DOI: https://doi.org/10.7554/eLife.47788.009

The following source data is available for figure 4:

**Source data 1.** FASTA formatted text file of mt*COI* sequence alignment.

DOI: https://doi.org/10.7554/eLife.47788.010

BTN2 allele (B). As in the mitochondrial control region, the closest published sequence to the *Mytilus* BTN2 allele is the sequence from 62mc10, a normal *M. trossulus* from the Baltic region. This again provides evidence that the two lineages of transmissible cancer arose from distinct *M. trossulus* individuals.

Unexpectedly, while the B allele was found in *Mytilus* BTN2 samples from both Europe and Argentina, it was not identified in the *M. chilensis* samples from Chile. In these two samples, two mt*COI* alleles were found, and they share a common allele that is likely to be the cancer-associated allele (Q). However, this allele is different from allele B, and it is *M. chilensis*-derived. These samples show evidence of the common *Mytilus* BTN2 alleles at all other nuclear and mitochondrial loci, so this suggests that there has been recombination between cancer and host mitochondria somewhere in the *M. chilensis* population, resulting in a subset of *Mytilus* BTN2 cells in Chile with new recombinant mitogenomes.

While two alleles (C and D) were identified in the *M. chilensis* mtCR region, only one allele was detected at mt*COI*. The control region alleles C and D were very similar, with the exception of a different recombination site with the male-like sequence, so it is likely that both mitogenomes have the same sequence at the more conserved *COI* locus. Additionally, in Castro26, a second recombinant sequence was identified that was composed of half *M. trossulus* sequence and half *M. chilensis* sequence. This could represent the second cancer mitogenome (i.e. mtCR D and mt*COI* Q could be on one mitogenome and mtCR C and the recombinant mt*COI* could be the other), but it was supported by a single clone, so it is unclear if it was a PCR artifact or a true second allele. Further analysis will be required to determine the full recombinant cancer mitogenomes in the *Mytilus* BTN2 cancers in this region of Chile.

## Detection of trans-oceanic cancer in multiple populations of *M. chilensis* and *M. edulis* using qPCR

The cancer-associated alleles identified at these four loci (*EF1α*, *H4*, mtCR, mt*COI*) were detected through cloning of PCR products from most of the *M. chilensis* and *M. edulis* individuals that were diagnosed as diseased, but not all (6 of 8, 5 of 8, 7 of 8, and 8 of 9, respectively). For example, the cancer-associated *EF1α* alleles were not cloned from *M. chilensis* sample Mch23. Quantitative PCR (qPCR) was used to determine whether the diseased animals in these cases, such as Mch23, had the cancer-associated alleles at a level too low to be identified through cloning or whether the transmissible cancer allele was absent from that individual. Primers specific for one of the cancer-associated *EF1α* alleles (H) were designed, and amplification was compared to amplification from 'universal' *Mytilus EF1α* primers (*Supplementary file 2*). Analysis of the *M. chilensis* samples showed that the cancer-associated allele was found in all three samples from Argentina that were diagnosed as diseased, including Mch23 (*Figure 5*), and it was undetectable in all three normal samples. In the *M. chilensis* from the Pacific Ocean, however, we found that two of the four diseased animals had high levels of the cancer-associated allele, but the other two did not. They likely have either conventional cancer or a different transmissible cancer lineage. Additional samples of all three *Mytilus* species were analyzed by this method. In *M. edulis*, qPCR showed the presence of the cancer-associated allele in all mussels diagnosed as diseased but not in normal samples. To test whether *Mytilus* BTN2 could be present in the Canadian individuals at a low level, qPCR analysis was conducted on the original samples from which *Mytilus* BTN1 was identified (*Metzger et al., 2016*). The allele associated with *Mytilus* BTN2 in *M. edulis* and *M. chilensis* was undetectable in both normal and diseased *M. trossulus* samples from Canada. qPCR primers specific for both cancer-associated *H4* alleles were made, and both were detected in all animals diagnosed as diseased except for the two Chilean samples that were negative for the cancer-associated *EF1α* allele (*Figure 5*). One allele was present at a higher level within each diseased animal than the other, but the ratio of the two alleles was not completely consistent across all individuals, suggesting that there has been recombination or copy number variation in some subgroups of the cancer lineage. The *H4* cancer-associated alleles were also identified in several normal *M. trossulus*. This confirms that *H4* alleles found in *Mytilus* BTN2 are present in some individuals in the normal *M. trossulus* population. This is expected, as each of the two transmissible cancer lineages arose in the past from a different normal *M. trossulus* individual.

Analysis of the two cancer-associated mitochondrial control region alleles shows that the C allele (only cloned from *M. chilensis* samples) is present in *M. chilensis* cancers at a very low level (<1% of total allele fraction). It was also undetectable in four of eight diseased *M. edulis*, again suggesting that this heteroplasmy may be lost in some of the cancer cells spreading in the *M. edulis* population (*Figure 5*).

Unexpectedly, when comparing the copy number of the D allele to amplification at a conserved locus about 200 bp away (shown schematically in *Figure 3B*), the neoplastic samples yielded values greater than 100%. Further investigation revealed a tandem repeat in the D mitochondrial genome, with multiple copies of the control region that are targeted by allele-specific primers. A tandem repeat was confirmed by generating the reverse complement of the qPCR primers specific for allele D and amplifying and sequencing the tandem repeat junction (marked in *Figure 3B*). While it cannot be directly used to estimate the percent of cancer cells in a sample, this locus is a highly sensitive marker of this lineage.

Analysis of the *mtCOI* locus by qPCR confirmed that the main *Mytilus* BTN2 allele was identified in all *M. edulis* cancer samples, and in the Argentinian *M. chilensis*, but not the Chilean *M. chilensis*.

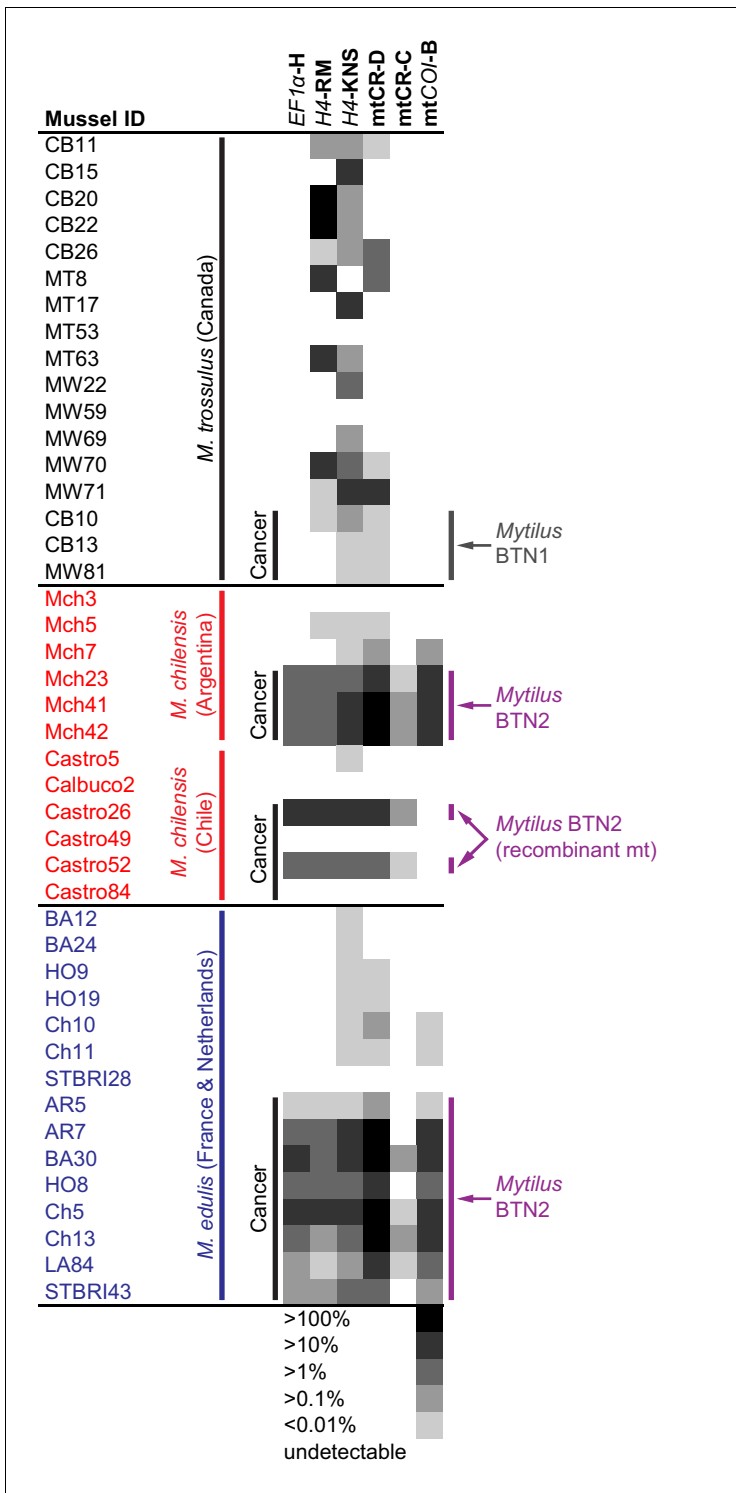

**Figure 5.** qPCR of multiple loci confirms the presence of the cancer lineage in *M. trossulus* and *M. edulis* from multiple populations. Allele-specific qPCR was conducted to determine the fraction of cancer-associated alleles at the four loci: allele H at *EF1α*, allele RM and KNS at *H4*, allele C and D at mtCR, and allele B at mt*COI* (see *Figure 5—source data 1*). Boxes show whether the cancer-specific allele is undetectable or detectible, with a heatmap showing the fraction of the total alleles at each locus that correspond to the specific allele (the quantity of amplification with an allele-specific qPCR reaction divided by the quantity amplified with 'universal' *Mytilus* primers). Each individual mussel is marked with its ID, species, and location, and each animal that was diagnosed as containing disseminated neoplasia is marked 'Cancer'.

*Figure 5 continued on next page*

*Figure 5 continued*

DOI: https://doi.org/10.7554/eLife.47788.011
The following source data is available for figure 5:
**Source data 1.** Tab separated text file with qPCR data (U means undetectable).
DOI: https://doi.org/10.7554/eLife.47788.012

This suggests that the mitochondria in the Chilean samples (Castro26 and 52) has been completely replaced with one that is a recombinant of the original *Mytilus* BTN2 *M. trossulus*-derived sequence and newly acquired *M. chilensis*-derived sequence.

Analyses at all loci confirm that two samples from Chile (Castro49 and 84) are not afflicted by *Mytilus* BTN2, while it is present at high levels in the other two diseased samples from Chile and all three tested samples from Argentina and eight from Europe. These two animals without any of the *Mytilus* BTN2 markers at any of the four loci thus do not have the *Mytilus* BTN2 cancer lineage. These data confirm that *Mytilus* BTN2 is distinct from *Mytilus* BTN1, and that it can be detected with multiple methods in both *M. chilensis* and *M. edulis* affected by disseminated neoplasia. Overall, the *EF1α* qPCR assay is the most highly sensitive and specific assay within these populations, with detection of the cancer-associated allele in all samples known to be affected by *Mytilus* BTN2 and no detectible amplification in any normal samples.

### Identification of *M. trossulus*-specific SNPs in *M. chilensis* and *M. edulis* samples

In order to confirm the presence of *M. trossulus*-derived cancer in *M. chilensis* and *M. edulis* individuals we tested for the presence of multiple *M. trossulus*-specific nuclear SNPs in *Mytilus* BTN2-positive specimens from Europe and Argentina, as well as additional samples of healthy mussels from those populations. Following previous studies (*Riquet et al., 2017*), we analysed SNP markers that are diagnostic between *M. trossulus* and either *M. edulis* or *M. chilensis*. We used 13 SNPs for each species. Five were used for both species, eight were specific to *M. edulis* hosts, and eight were specific to *M. chilensis* (*Supplementary file 3*). We confirm that all six *Mytilus* BTN2-positive *M. edulis* and all three positive *M. chilensis* analysed with the SNP assay have strong fluorescence corresponding to the *M. trossulus* allele at all diagnostic loci analysed and they are clear outliers in the compound multi-locus average (*Figure 6*). In contrast, no normal animals of either species have a signal of an *M. trossulus* allele at more than one locus. Therefore, data from multiple independent loci provide further evidence supporting the finding that the *Mytilus* BTN2 lineage is an *M. trossulus* derived cancer.

## Discussion

Disseminated neoplasia has been previously shown to be due to a transmissible cancer lineage in four bivalve species, including *Mytilus trossulus* (*Metzger et al., 2016*; *Metzger et al., 2015*), and the current study greatly extends both our understanding of cross-species transmission of cancer (from *M. trossulus* into two other *Mytilus* species) and our understanding of the potential for geographic spread of transmissible cancers (*Figure 7*). This finding of a single lineage crossing into multiple species across such vast distances is unexpected. Phylogenetic evidence from four loci and SNP data from 13 diagnostic nuclear SNPs show that the lineage identified here (*Mytilus* BTN2) is distinct from *Mytilus* BTN1, and that it too arose from a *M. trossulus* individual before crossing into other *Mytilus* species, as previously proposed (*Riquet et al., 2017*). The gene trees of the four loci are all concordant with each other, and are all consistent with a clonal, asexual cancer lineage derived from an *M. trossulus* cancer that is independent from *Mytilus* BTN1. The only exception to this is the observation of recombination with host mitochondrial DNA in the mitogenomes of *Mytilus* BTN2 in Chile. The age of this cancer lineage is unknown, but the other transmissible cancer with world-wide spread (canine transmissible venereal tumor; CTVT) was estimated to be thousands of years old (*Murchison et al., 2014*; *Baez-Ortega et al., 2019*). Our oldest samples are 10 years old, but if the earliest reports of disseminated neoplasia in *M. edulis* were attributable to this same lineage (*Farley, 1969*), it would be at least 50 years old.

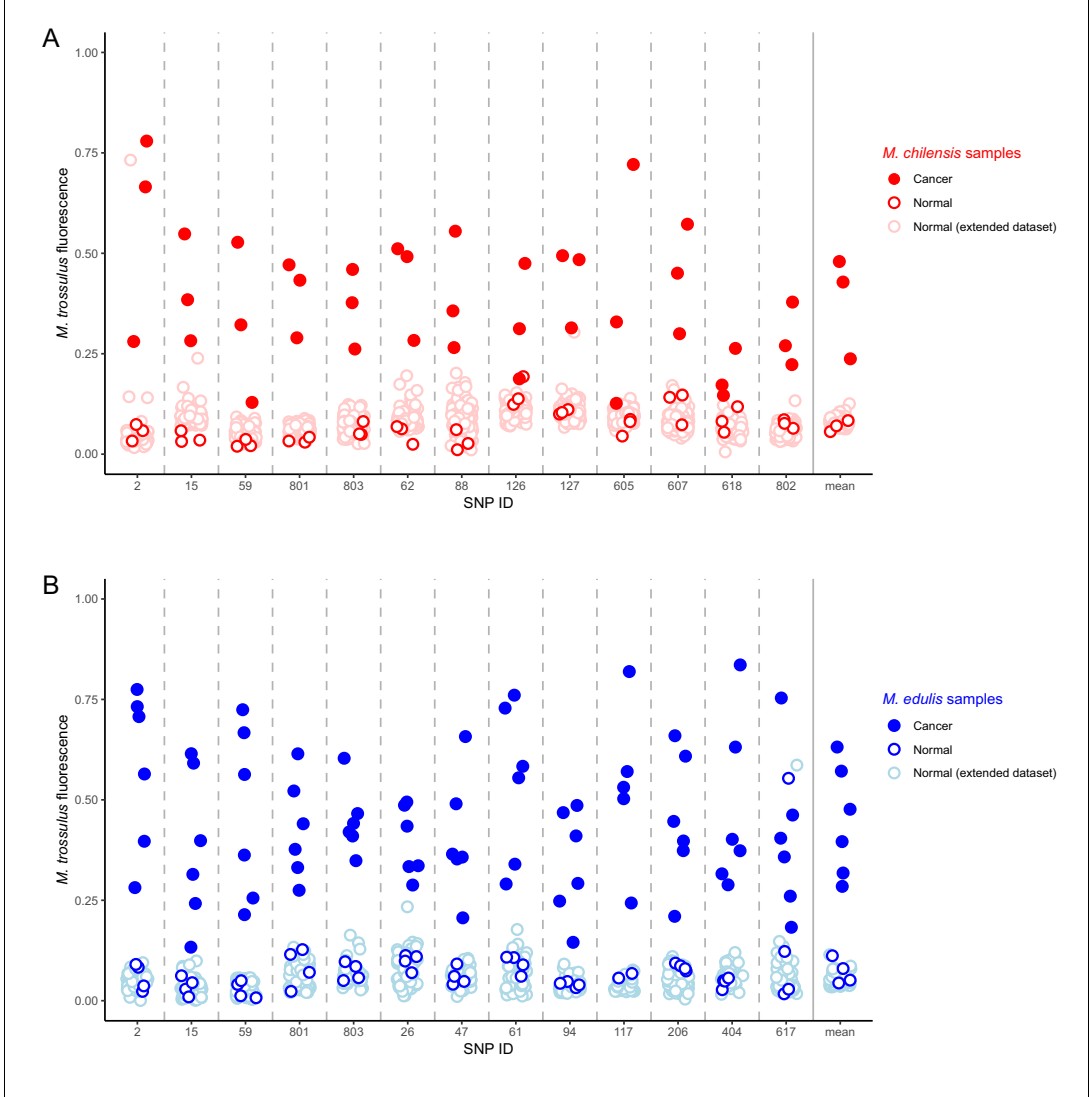

**Figure 6.** Identification of *M. trossulus*-specific SNPs in *M. chilensis* and *M. edulis* samples. Cancer and normal samples of (**A**) *M. chilensis* from Argentina and (**B**) *M. edulis* from Europe were analyzed for the presence of 13 SNPs that are diagnostic for *M. trossulus* alleles. The fraction of the fluorescence attributable to the *M. trossulus* allele is shown for each SNP, and the mean multi-locus estimate is shown on the right. SNP IDs indicate shortened names of the specific nuclear SNP loci used (see ***Supplementary file 3***). Red points indicate *M. chilensis* and blue indicate *M. edulis*. Dark filled circles represent cancer samples, dark open circles represent normal samples which were a part of the sequence and qPCR analysis (***Figures 1–4***), and light open circles represent samples from an extended set of normal individuals from the same populations. Cancer samples included: Mch23, Mch41, and Mch42 for *M. chilensis*, and BA30, HO8, Ch5, Ch13, LA84, and STBRI43 from *M. edulis*. Normal samples included Mch3, Mch5, and Mch7 for *M. chilensis*, and HO9, HO19, Ch10, and Ch11 for *M. edulis*, as well as 109 additional normal *M. chilensis* and 62 additional normal *M. edulis*. See ***Figure 6—source data 1*** for *M. chilensis* data and ***Figure 6—source data 2*** for *M. edulis* data.

DOI: https://doi.org/10.7554/eLife.47788.013

The following source data is available for figure 6:

**Source data 1.** Comma separated text file with SNP data from *M. chilensis* samples.
DOI: https://doi.org/10.7554/eLife.47788.014

**Source data 2.** Comma separated text file with SNP data from *M. edulis* samples.
DOI: https://doi.org/10.7554/eLife.47788.015

The transmission of cancer cells from the Northern to the Southern Hemisphere and from Atlantic to Pacific Oceans is remarkable, and ocean currents are unlikely to provide a mechanism for travel in either direction (especially given that anti-tropical distribution prevents a stepping-stone for trans-equatorial propagation). The most likely route of transmission is human intervention, specifically the

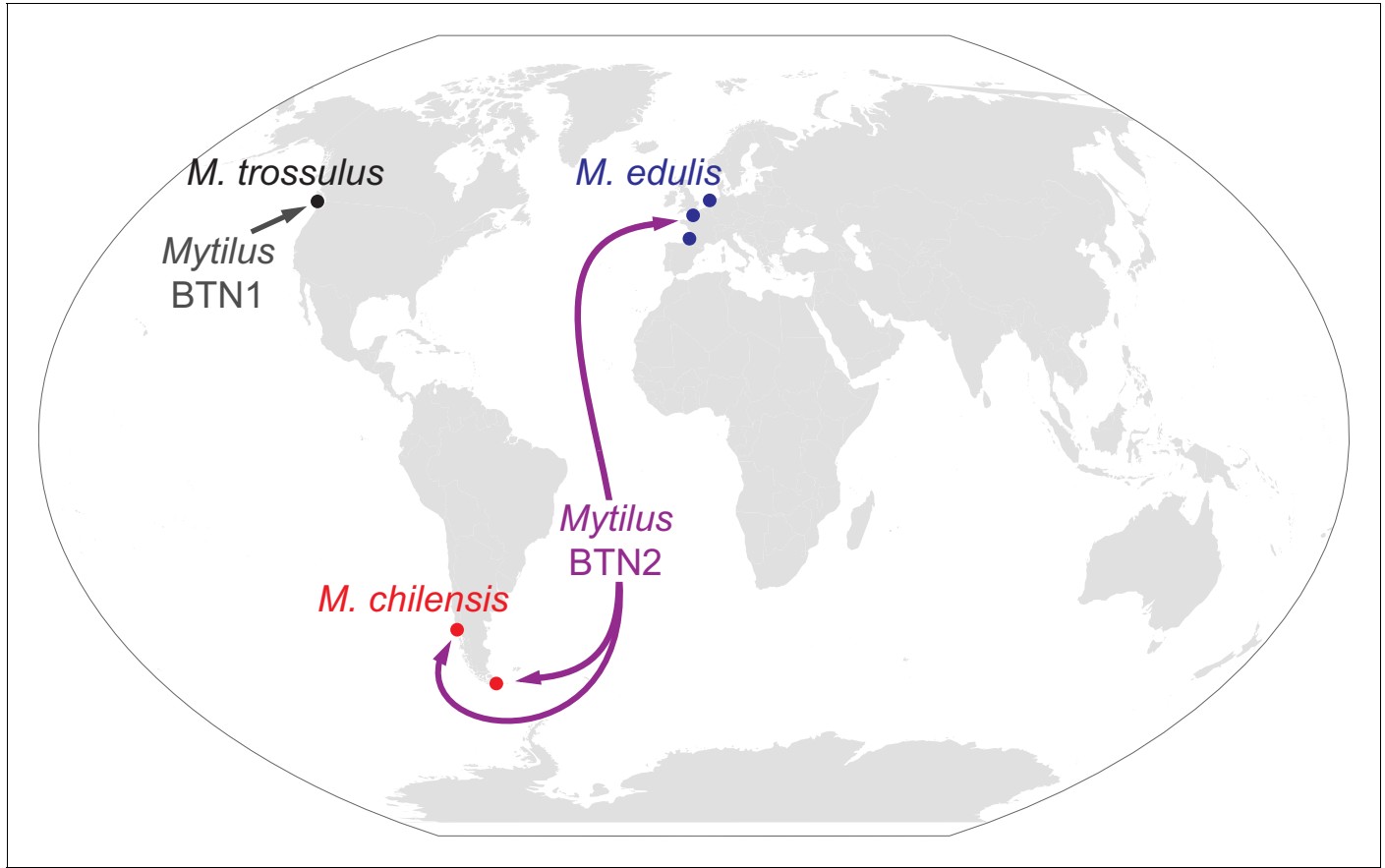

**Figure 7.** Map of the spread of *Mytilus* BTN1 and 2. The locations of all collections in the current study are shown with circles. Diseased animals were found at all collection locations. The species collected at each location is marked. The *Mytilus* BTN1 lineage (gray) was only identified in *M. trossulus* populations in BC, Canada, while the *Mytilus* BTN2 lineage (purple) was identified in both *M. edulis* and *M. chilensis* populations in both the Pacific and Atlantic Oceans. Both cancer lineages originated in an *M. trossulus* individual, although the exact locations of the origins are unknown. The direction of spread of these cancers is unknown. Winkel-Tripel map was created by Flappiefh (Wikimedia Commons) using gringer's perlshaper script, and Natural Earth (naturalearthdata.com).

DOI: https://doi.org/10.7554/eLife.47788.016

transportation of animals harboring neoplastic cells in or on shipping vessels. While it is difficult to experimentally prove that a particular disease transmission was due to accidental transport on shipping vessels, examples of anthropogenic introduction of invasive species via shipping transport and aquaculture are common throughout the globe (*Molnar et al., 2008*). The accidental anthropogenic introduction of *Mytilus* species includes *M. galloprovincialis* introduced in northern Chile (*Daguin and Borsa, 2000*) and Argentina (*Zbawicka et al., 2018*), *M. edulis* introduced in BC (*Crego-Prieto et al., 2015*) and the Kerguelen Islands (*Fraisse et al., 2018*), and *M. trossulus* into Scotland (*Dias et al., 2009*). Since *Mytilus* species are known to adhere to ships and spread accidentally across the world, and transmissible cancer can have an incubation period of weeks to months, any diseased animal which adheres to a ship could lead to the spread of a transmissible cancer to new and potentially susceptible populations. If transmissible cancers are able to readily travel along shipping lines and infect closely related species, then this and other lineages of transmissible cancer have the potential for world-wide reach.

When identifying identical or nearly identical sequences in samples from different locations, care must be taken to prevent a small amount of contamination from leading to a false positive. In this case, the initial sequencing and amplification of the $EF1\alpha$ loci from *M. edulis* and *M. chilensis* samples were conducted by different people at different sites before any plasmids containing the sequence from one site had been transferred to the other. The $EF1\alpha$ qPCR assay was independently replicated on European *M. edulis* infected individuals in two labs, in USA and France. Also, the

finding of slight differences between the two subgroups of this cancer lineage argue against a single source of contamination. Additionally, qPCR verification confirms that a significant fraction of the DNA of each sample contains the cancer-associated allele, and that it is not found in DNA samples from normal animals, which have been treated in the same way. Therefore, the data could not be due to contamination with a small amount of DNA and represents genuine identification of a nearly identical cancer from mussels of different continents and species.

Invertebrates do not possess a vertebrate-like adaptive immune system or a major histocompatibility complex, which vertebrates utilize for discriminating self from nonself. However, in most previously identified cases of BTN, the neoplasia has only been identified in the species of origin (*Metzger et al., 2016*; *Metzger et al., 2015*), and attempts to experimentally transfer neoplastic cells from *M. trossulus* or the soft-shell clam *Mya arenaria* into several bivalve species have only been successful when the transfer is between members of the same species (*Kent et al., 1991*; *Mateo et al., 2016*). The species in the *Mytilus* complex are able to hybridize, so the barrier to cross-species transmission may be relatively low. However, it has been reported that the disseminated neoplasia found on the Pacific coast of North America (which was identified as originating from *M. trossulus*) was found in *M. trossulus* at a much higher prevalence than *M. edulis* in the same region (*Muttray et al., 2007*), which suggested that the *M. trossulus* cancer preferentially infected *M. trossulus* hosts. It is currently unknown whether *Mytilus* BTN2 would be able to infect its original host species or if the original host has evolved resistance, as was suggested with the cancer derived from the pullet shell clam *Venerupis corrugata* (*Metzger et al., 2016*). *M. chilensis* is closely related to *M. edulis*, and it has even been suggested that it could be considered a subspecies, *M. edulis platensis* (*Borsa et al., 2012*), although this is controversial (*Zbawicka et al., 2018*), so once the cancer was able to cross into either *M. chilensis* or *M. edulis*, the second jump into the other species may have presented a lower barrier. Based on analysis of transmissible cancers in devils and dogs it has been hypothesized that transmissible cancers primarily occur in inbred populations, but the finding of this cancer spreading into highly outbred individuals from multiple different species within the same genus shows that this constraint is not universal.

The evidence for mitochondrial heteroplasmy and recombination observed in this cancer lineage and its loss or gain is a very interesting observation that suggests a dynamic mitochondrial population in BTN, which may not exactly match the nuclear evolutionary path. It has been shown that mitochondria in CTVT has been replaced several times by mitochondrial capture of new normal mitochondrial genomes (*Rebbeck et al., 2011*; *Strakova et al., 2016*), so a dynamic mitochondrial population may be a common feature among transmissible cancers. It is also possible that one of the two cancer-associated mtCR regions could be a nuclear-to-mitochondrial transfer (numt) of a copy of the control region instead of mitochondrial heteroplasmy, as this possibility cannot be excluded by PCR of total extracted DNA. If this were the case, the sequences would still be evidence of an *M. trossulus*-derived transmissible cancer that is distinct from *Mytilus* BTN1. Mitochondrial replacement has not yet been described in any BTN, so further analysis will be required to determine the full recombinant cancer mitogenomes and to test whether there is any selective advantage of this recombinant mitogenome in *Mytilus* BTN2.

Comparison of cancer-associated sequences to those found in normal *M. trossulus* populations may allow us to identify the location of the origin of these cancer lineages. BLAST searches for the neoplasia-associated alleles most closely match *M. trossulus* from the Baltic and BC. The *Mytilus* BTN2 recombinant mitochondrial sequences closely match the recombinant sequence first identified in the Baltic (62mc10), but this haplotype is actually rare in the Baltic, with similar recombinant haplotypes more commonly found on the coast of Norway (*Śmietanka and Burzyński, 2017*). This suggests that this cancer lineage originated in Northern Europe, but it may not necessarily be from the Baltic. However, there are populations of *M. trossulus* in many locations around the world, and not all have been sequenced at the loci analyzed here. Additionally, the mtCR C and D alleles may be directly related to 62mc10, or they may represent novel recombinations in the same region (there is a single base difference in the recombination region between the D allele and 62mc10 which suggests that the recombination points are not identical). Multiple recombination events in which male sequence has recombined into the female mitochondrial lineage in that region (including several cases with tandem duplications) have been identified in *M. trossulus* (*Burzyński et al., 2006*). Additionally, while exact matches of a *Mytilus* BTN2 *H4* allele was identified in *M. trossulus* from BC,

exact matches of the alleles at other loci have not been identified in any wild *M. trossulus* populations. Therefore, the exact population of origin of this cancer remains uncertain.

Overall, these data show that two lineages of cancer from two separate *M. trossulus* individuals have turned into BTN lineages, and one of these lineages (*Mytilus* BTN2) has spread world-wide throughout multiple host species. This spread also suggests that anthropogenic influence may exacerbate the spread of cancers and may allow access into new environments and new populations, where the pathogenic potential is unknown.

## Materials and methods

### *M. chilensis* collection and diagnosis

60 individual market-sized mussels (mean, 67.6 mm; range, 52–92 mm in the longest axis) were collected in February 2012 from a culture at Bahía Brown (54°52′S, 67°31′W), Beagle Channel, Argentina. The soft parts of the specimens were carefully removed from their shells and fixed in Davidson's solution for 24 hr. Oblique transverse sections, approximately 5 mm thick, including mantle, gills, gonad, digestive gland, nephridia, and foot were taken from each specimen. Tissue samples were embedded in paraffin, and 5 µm sections were stained with hematoxylin and eosin. Histological sections were examined using a Leica DM 2500 light microscope for the presence of neoplastic cells. Small samples of gill from each mussel were preserved in 100% ethanol for DNA extraction.

200 individuals were collected from farmed populations at two sites in Chile (Calbuco and Castro). Hemolymph was drawn from adductor muscle, centrifuged at low speed and the hemocytes were stored in RNAlater (ThermoFisher Scientific) for DNA extraction. The animals were opened and the full body with one valve were fixed for at least one week with 10% formaldehyde in PBS. Tissue samples were embedded in paraffin, sectioned and stained as above. The animals were diagnosed as diseased if infiltration of hemocytes with a high nucleus-cytoplasm ratio and granular chromatin were observed.

Healthy mussels from other populations of *M. chilensis* around Punta Arenas, Chile (n = 46) and from islands around Parque Nacional Alberto de Agostini, Chile (n = 63) were analyzed with the SNP assay.

### *M. edulis* collection and diagnosis

We used a collection of samples (gills in 90% ethanol) accumulated by the ISEM lab for years in their study of hybridization and introgression in the *M. edulis* complex of species. *M. trossulus* is not naturally found in France and the genetic analysis of thousands of individuals did not identify any *M. trossulus* individuals even in ports or mussel farms (*Simon et al., 2019*). As a consequence, a signal of amplification of trossulus alleles at several diagnostic SNPs was used as a diagnosis for mussels likely infected by a *M. trossulus* BTN (*Riquet et al., 2017*). Here we first used specimens identified by Riquet et al.: two mussels sampled in the Arcachon Bay (South-West of France) in 2016 (AR5 and AR7), one mussel sampled in Barfleur (Normandy, France) in 2015 (BA30), and one mussel sampled in the Wadden Sea (The Netherlands) in 2009 (HO8). To these four mussels already described, we added two additional mussels identified in a new extensive genetic survey of >4000 individuals (*Simon et al., 2019*). These two mussels were collected from Chausey Island (English Channel) in 2009 (Ch5 and Ch13). For comparison, two healthy individuals were randomly chosen from each of the collections from Chausey, Barfleur, and the Wadden Sea (Ch10, Ch11, BA12, BA24, HO9, HO19). Additional healthy mussels from the populations of Chausey (n = 9), Barfleur (n = 28) and Wadden Sea (n = 21) were analyzed with the SNP assay.

In parallel, a hemocytological analysis was undertaken to identify BTN infected mussels on approximately one hundred individuals from Brittany. Briefly, the mussels were anaesthetized by bathing them for 30 min in a solution containing 50 g/L of magnesium chloride dissolved in two-thirds of distilled water and one-third of sea water (*Suquet et al., 2009*). 40 µl of hemolymph were withdrawn from the adductor muscle with a sterile 1 mL syringe fitted with a 27-gauge needle. After the addition of 160 µl of artificial seawater, the hemolymph solution was 'cytospun' onto coated slides (Shandon Cytospin four and Cytoslides, Thermo Scientific; 10 min, 800 rpm), fixed, stained with May-Grünwald Giemsa, and observed under light microscopy. The whole area of the cell spot

on the cytoslide was observed in all the samples in search of neoplastic cells, characterized by an enlarged diameter, elevated nucleus-cytoplasm ratio, and high basophily. Two individuals that also showed a signal of *M. trossulus* allele amplification from gill tissue DNA were chosen for the present study. One was sampled in Lannion (LA84) and the other in Saint Brieux (STBRI43) in 2017. An additional normal individual from Saint Brieux (STBRI28) was also analyzed as a control, and additional healthy individuals from Lannion (n = 4) were analyzed with the SNP assay.

## DNA extraction
DNA was extracted from ethanol-fixed gill and siphon samples using DNeasy Blood and Tissue Kit (Qiagen) with the following additional steps to remove PCR-inhibiting polysaccharides. After tissue lysis, 65 µl of P3 Buffer (Qiagen) was added for 5 min to precipitate out polysaccharides and spun down at 17k × g at 4°C for 5 min. The resulting supernatant was transferred to a new tube, 200 µl of Buffer AL was added, and the manufacturer's protocol resumed. DNA was extracted from hemocyte samples using the standard DNeasy Blood and Tissue Kit protocol.

## PCR and cloning
Primers and annealing temperatures are listed in *Supplementary file 1*. PCR amplification at the $EF1\alpha$ locus was done using PfuUltra II Fusion HS DNA Polymerase (Agilent) with a 15 s extension time. Amplification at the *H4,* mtCR, and mtCOI loci were achieved using Q5 Hot Start High-Fidelity DNA Polymerase (NEB) with a 30 s extension time. Primers for *H4* were designed to span the full *H4* gene (*Eirín-López et al., 2004*). Primers for mtCR were adapted from Burzynski et al. and were designed to cover lrRNA, a variable domain, and a conserved domain in the mitochondrial DNA control region (*Burzyński et al., 2006*). 'Pan-molluscan' barcoding primers (*Folmer et al., 1994*) did not amplify the *Mytilus* BTN2 mtCOI allele, so degenerate primers flanking the standard mtCOI were used. In all cases, 25–50 ng of genomic DNA was amplified for 35 cycles.

PCR products were gel extracted using spin columns (Qiagen, NEB) and either directly sequenced, or, when multiple alleles at a locus could not be resolved by direct sequencing, were cloned using the Zero Blunt TOPO PCR Cloning Kit (Invitrogen). Plasmids were transformed into TOP10 or DH5$\alpha$ competent *E. coli* (Invitrogen) and 6–12 clones were picked for further sequencing using M13F and M13R primers (Genewiz). The primer binding regions were excluded from sequence analysis, and all unique alleles were identified. In cases where a sequence was found in only a single clone and that clone differed by <0.5% (two differences for $EF1\alpha$ and *H4*, three for mtCR) from another allele that was supported by more than one clone from the same individual, the sequence with higher support was used as the correct allele. Additionally, a sequence which was found in only one clone and which is consistent with a recombination event between two alleles from the same sample was discarded. For $EF1\alpha$, every allele presented was found in at least two clones (two sequences found only in single clones were removed from analysis). Alleles were named arbitrarily, such that each allele name (A-2H) represents the same sequence at that locus.

## Phylogenetic analysis
Coding sequences had minimal indels and were aligned manually. The mtCR locus was aligned with MUSCLE 3.8.31 (*Edgar, 2004*) with minor manual adjustment. Maximum likelihood phylogenetic trees were generated using PhyML 3.0 (*Guindon et al., 2010*), with 100 bootstrap replicates, which treats gaps in the alignment as missing data, with automatic model selection through Akaike Information Criterion. Trees were visualized using FigTree version 1.4.4. All alignments are included as source data, and all sequences available in GenBank (accession numbers: $EF1\alpha$, MN546736-MN546756; *H4*, MN546757 - MN546792; mtCR, MN546793 - MN546830; mtCOI, MN546831 - MN546858).

## qPCR
Cancer-associated allele-specific qPCR was performed in *M. trossulus*, *M. edulis*, and *M. chilensis* genomic DNA samples using PowerUp SYBR Green Master Mix (Applied Biosystems). For all targets, one primer set was designed to specifically amplify a cancer-associated allele for a given locus ($EF1\alpha$: Allele H; *H4*: Allele RM and KNS; mtCR: Allele C and D: mtCOI: Allele B). A second pair of universal control primers were designed to amplify all *Mytilus* alleles at each locus. Primers are listed

in *Supplementary file 2*. Standards for all targets consisted of a single plasmid containing the relevant cancer-associated allele target, which was linearized by NotI-HF (NEB) restriction digest before qPCR. Serial dilutions ($10^7$–$10^1$ copies) of linearized standard plasmid in triplicate were used to generate a standard curve for each qPCR run. In all experimental cases, 10–100 ng of each genomic DNA sample was run in triplicate 10 µl reactions.

All qPCR assays were performed using the Applied Biosystems StepOnePlus Real-Time PCR System (Applied Biosystems) and associated software. Allele-specific qPCR at the *EF1α* locus was comprised of a 3-step cycling stage (95℃ for 15 s, 50℃ for 15 s, and 60℃ for 30 s, respectively) for 40 cycles, whereas qPCR at the *H4,* mtCR, and mt*COI* loci were run using a 2-step cycling stage protocol (95℃ for 15 s, 60℃ for 30 s) for 40 cycles. A melting curve was generated to determine amplification specificity according to the following parameters: an initial denaturation step at 95℃ for 15 s, followed by a gradual temperature increase from 60℃ to 95℃ (Ramp of +0.3℃). The ratio of the cancer-associated allele to all alleles present at a given locus was calculated for each sample using the average copy number amplified by cancer-associated primers in triplicate reactions divided by the average copy number amplified by the universal primers in triplicate reactions. For each sample, the limit of detection for the allele-specific qPCR was set at 10 copies/reaction, and DNA samples were discarded if the universal *Mytilus* reaction at any of the loci yielded <5000 copies/reaction (upper limit of detection of 0.2%).

## SNP array

The SNPs used were previously described (*Simon et al., 2018*). They were identified as being diagnostic (nearly fixed for alternative alleles) in a large sequence dataset (*Fraïsse et al., 2016*) and were subsequently verified to remain diagnostic with larger sample sizes (*Simon et al., 2019*). Non-hybrid individuals can nonetheless occasionally be heterozygous at one, or very rarely at two, of these diagnostic loci because incomplete lineage sorting and introgression is pervasive in the *M. edulis* complex of species (*Fraïsse et al., 2018*), but never at more than two loci. Genotyping was subcontracted to LGC-genomics and performed with the Kompetitive Allele Specific PCR (KASP) chemistry (*Semagn et al., 2014*). The results are a combination of two fluorescence values, one for allele 1 (x), and another for allele 2 (y), the *M. trossulus*-specific allele. Following a method used for genotyping polyploids (*Cuenca et al., 2013*), we transformed the data to have a single measure of the relative fluorescence of the two alleles, from 0 when the fluorescence of allele one dominates, to one when the fluorescence of allele 2 (*i.e.* the *M. trossulus* allele) dominates, using the following formula: $y'=y/(x+y)$. The fluorescence of the *M. trossulus* alleles was averaged over all 13 diagnostic loci for each species to obtain a compound multi-locus estimate (see *Supplementary file 3*).

## Acknowledgements

The authors express their gratitude to Lic. Andrés Fernández from Secretaría de Desarrollo Sustentable y Ambiente of Tierra del Fuego province for sampling activities. We thank J Ausió for help in collection of *M. trossulus* from Vancouver Island, Canada, and Jorge Gomez for help in the collection of *M. chilensis* from Calbuco and Castro, Chile.

## Additional information

### Funding

| Funder | Grant reference number | Author |
| --- | --- | --- |
| National Institutes of Health | K22 CA226047 | Michael J Metzger |
| Howard Hughes Medical Institute | Goff Lab | Stephen P Goff |
| Howard Hughes Medical Institute | EXROP | Maria Polo-Prieto |
| Fondo Nacional de Desarrollo Científico y Tecnológico | FONDECYT1180705 | Gloria Arriagada |

| Montpellier Université d'Excellence | BLUECANCER | Maurine Hammel<br>Alexis Simon<br>Nicolas Bierne |
| --- | --- | --- |
| Agence Nationale de la Recherche | ANR-18-CE35-0009 | Maurine Hammel<br>Alexis Simon<br>Nicolas Bierne |
| Consejo Nacional de Investigaciones Científicas y Técnicas | 2014-0670 | Nuria Vázquez |

The funders had no role in study design, data collection and interpretation, or the decision to submit the work for publication.

### Author contributions

Marisa A Yonemitsu, Data curation, Investigation, Writing—review and editing; Rachael M Giersch, Maria Polo-Prieto, Alexis Simon, Florencia Cremonte, Fernando T Avilés, Nicolás Merino-Véliz, Investigation, Writing—review and editing; Maurine Hammel, Erika AV Burioli, Resources, Supervision, Investigation, Writing—review and editing; Annette F Muttray, James Sherry, Carol Reinisch, Susan A Baldwin, Resources, Writing—review and editing; Stephen P Goff, Conceptualization, Resources, Supervision, Project administration, Writing—review and editing; Maryline Houssin, Gloria Arriagada, Supervision, Investigation, Writing—review and editing; Nuria Vázquez, Nicolas Bierne, Conceptualization, Supervision, Investigation, Writing—review and editing; Michael J Metzger, Conceptualization, Data curation, Supervision, Funding acquisition, Validation, Investigation, Visualization, Methodology, Writing—original draft, Project administration, Writing—review and editing

### Author ORCIDs

Marisa A Yonemitsu (iD) https://orcid.org/0000-0003-3290-4306
Alexis Simon (iD) http://orcid.org/0000-0002-6176-5045
Stephen P Goff (iD) http://orcid.org/0000-0002-9679-0582
Nicolas Bierne (iD) https://orcid.org/0000-0003-1856-3197
Michael J Metzger (iD) https://orcid.org/0000-0001-7855-1388

### Decision letter and Author response

Decision letter https://doi.org/10.7554/eLife.47788.024
Author response https://doi.org/10.7554/eLife.47788.025

# Additional files

### Supplementary files

• Supplementary file 1. Table of PCR primers used for amplification and sequencing.
DOI: https://doi.org/10.7554/eLife.47788.017

• Supplementary file 2. Table of qPCR primers used.
DOI: https://doi.org/10.7554/eLife.47788.018

• Supplementary file 3. Table of diagnostic SNP loci used to discriminate *M. trossulus* from *M. edulis* and *M. chilensis.*
DOI: https://doi.org/10.7554/eLife.47788.019

• Transparent reporting form DOI: https://doi.org/10.7554/eLife.47788.020

### Data availability

All data generated or analyzed during this study are included in the manuscript and supporting files. Source data files have been provided for Figures 1-5 and S1. Sequences are available in BioProject (PRJNA580289).

The following dataset was generated:

| Author(s) | Year | Dataset title | Dataset URL | Database and Identifier |
|---|---|---|---|---|
| Marisa A Yonemitsu, Rachael M Giersch, Maria Polo-Prieto, Maurine Hammel, Alexis Simon, Florencia Cremonte, Fernando T Avilés, Nicolás Merino-Véliz, Erika AV Burioli, Annette F Muttray, James Sherry, Carol Reinisch, Susan A Baldwin, Stephen P Goff, Maryline Houssin, Gloria Arriagada, Nuria Vázquez, Nicolas Bierne, Michael J Metzger | 2019 | Mytilus trossulus, Mytilus chilensis, Mytilus edulis Variation | https://www.ncbi.nlm.nih.gov/bioproject/?term=PRJNA580289 | NCBI BioProject, PRJNA580289 |

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
