## [Decision Letter]

Thank you for submitting your article "A single clonal lineage of transmissible cancer identified in two marine mussel species in South America and Europe" for consideration by *eLife*. Your article has been reviewed by three peer reviewers, one of whom served as guest Reviewing Editor, and the evaluation has been overseen by Patricia Wittkopp as the Senior Editor. The reviewers have opted to remain anonymous.

The reviewers have discussed the reviews with one another and the Reviewing Editor has drafted this decision to help you prepare a revised submission.

Summary:

The identification of an independent origin of a transmissible cancer has the potential to impact the field significantly. This can inform on the likelihood of neoplastic transformation and transmission. In this paper the authors argue that an infectious cancer in two mussel species likely originated in a third species. Bivalve neoplasia had been previously documented in this third species (*M. trossulus*), but the authors claim that the cancer from this study differs from the previously identified cancer in *M. trossulus*. The authors are making two claims: (1) BTN2 is distinct from BTN1, and (2) BTN2 has spread from the Pacific coast of South America to the Atlantic coast of Europe.

Essential Revisions:

1) Sequencing additional loci is needed to definitively show that BTN2 in Chile/Argentina is in fact the same cancer strain found in Europe. Also, sequencing less than 1kb over 3 loci makes drawing conclusions about origin and hybridization tricky. As example, how did they determine that the EF1a 'cancer' alleles were not copy number variants in the host? Were noncancerous tissues tested for absence of these alleles in carrier individuals? Finally, with such small sequencing, how are the authors certain they are not sequencing numts, which are common in numerous marine invertebrate species. Additional sequencing will be needed to respond to this.

2) Gene trees and species trees are often discordant, and the authors have not sampled enough loci and enough of the coalescent history to resolve this discordance. Gene discordance across the genome is now more the rule than the exception. The single coalescent history of the asexual BTN lineage should help ameliorate some of this, but more than three markers are needed. Also, in the same vein, the authors themselves highlight, their "tree" is incompatible with the G and H alleles coming from BTN1S. But this is based on one locus, which is insufficient.

Issues that affect interpretation of the data: response should either include significant clarification or rewriting of sections:

1) What is meant by "a fraction" of *M. trossulus*" SNPs? How many? Might this be sequencing error?

2) What is meant by: We sequenced…and found more than the normal two alleles in "most" of them…What is "most"? There are many such examples.

3) The qPCR results from attempted cloning of disease alleles is confusing. Finding that two of four disease *M. chilensis* animals have high levels of the cancer associated allele while the other two did not – are the authors suggesting yet another transmissible cancer lineage?

4) Given the inability to consistently amplify cancer alleles in various samples, it is unclear how any conclusions regarding the contribution of the Canadian samples and *M. trossulus* can be made.

5) It is also unclear how to interpret the fact that H4 cancer associated alleles are found in several normal *M. trossulus* animals. How many is "some?"

6) Sequencing less than 1kb over 3 loci makes drawing conclusions about origin and hybridization tricky. As example, how did they determine that the EF1a 'cancer' alleles were not copy number variants in the host? Were noncancerous tissues tested for absence of these alleles in carrier individuals? Specifically, looking only at normal alleles, there is a clear grouping based on species of origin in a phylogenetic tree There are three exceptions to this clustering, from one normal *M. trossulus* (MW59) and two diseased *M. chilensis* (Mch41 and Castro26) with some distinctly *M. edulis*-like alleles. These may reflect introgression or true hybrid individuals-MW59 also has an EF1α allele corresponding to *M. edulis* and likely represents a hybrid with this species, which has been introduced into Canada (Crego-Prieto et al., 2015)." This sort of noise, although expected, becomes problematic when it encompasses a third of the data. Please explain in more detail your rationale for your conclusions and include a discussion of caveats.

---

## [Author Response]

Essential Revisions:1) Sequencing additional loci is needed to definitively show that BTN2 in Chile/Argentina is in fact the same cancer strain found in Europe. Also, sequencing less than 1kb over 3 loci makes drawing conclusions about origin and hybridization tricky. As example, how did they determine that the EF1a 'cancer' alleles were not copy number variants in the host? Were noncancerous tissues tested for absence of these alleles in carrier individuals? Finally, with such small sequencing, how are the authors certain they are not sequencing numts, which are common in numerous marine invertebrate species. Additional sequencing will be needed to respond to this.

We have now included additional sequence data which support these claims. We have sequence data from an additional mitochondrial locus (mt*COI*), which confirms that alleles from *M. edulis* and *M. chilensis* are identical and derived from *M. trossulus*, and that they are distinct from *Mytilus* BTN1 (Figure 4). We did find that two samples from Chile showed evidence of recombination (the only evidence of recombination detected so far across all loci tested). We also used a SNP array on cancerous mussels as well as a large sample of healthy individuals to identify the presence of SNPs that are unique to *M. trossulus* (Figure 6). This allowed us to look at many independent nuclear loci to confirm that the cancer cells are *M. trossulus* in origin, and that this signal of *M. trossulus* SNPs could not be found in any normal animals from *M. chilensis* or *M. edulis* populations.

Regarding the question of whether the additional alleles could be simply copy number variation, the reviewers are correct that the finding of multiple alleles alone is not sufficient to support the claim of a transmissible cancer – the key finding is that there are multiple alleles and that some of those are cancer-associated alleles that are identical between independent individuals. Genome duplication within an *M. chilensis* individual, for example, would be expected to give rise to multiple alleles phylogenetically related to normal *M. chilensis* alleles. It would not be expected to give rise to a second allele which is nearly identical to that of a different species. And it would not be expected to generate exactly the same sequence in multiple individuals from two different species.

Additionally, we saw no evidence of multiple alleles in any normal samples. When conducting PCR and sequencing from total DNA it is not possible to distinguish a true mitochondrial gene from a nuclear-to-mitochondrial transfer (numt). The reviewers are correct that we cannot rule this out, and we have clarified this in the text. However, if one of the cancer associated “mtCR” alleles were to be a numt, for example, it would not change the conclusion that there is sequence derived from *M. trossulus* mitochondrial DNA that is nearly identical in cancers across multiple individuals in two species.

2) Gene trees and species trees are often discordant, and the authors have not sampled enough loci and enough of the coalescent history to resolve this discordance. Gene discordance across the genome is now more the rule than the exception. The single coalescent history of the asexual BTN lineage should help ameliorate some of this, but more than three markers are needed. Also, in the same vein, the authors themselves highlight, their "tree" is incompatible with the G and H alleles coming from BTN1S. But this is based on one locus, which is insufficient.

Gene trees and species trees are often discordant, but as the reviewers mention, a clonal transmissible cancer is asexual, so we do not expect to see discordance between a gene tree and a cancer lineage tree except in the case of recombination. We do see some discordance in the normal sexual mussel samples, as expected (i.e. *M. chilensis* has introgressed *M. trossulus* sequence at the *EF1alpha* locus and a small number of individuals appear to be hybrids), but we see no discordance between any of the gene trees in any of the four loci, with the exception of the mitochondrial recombination at one locus in two Chilean samples. Also, at all loci, the gene trees support a clonal expansion of a nearly identical asexual lineage derived from *M. trossulus* and distinct from *Mytilus* BTN1. We had specifically highlighted the evidence that neither allele G nor H were likely to have a common cancer ancestor with allele S, as noted, but we had not specifically mentioned that each other locus provides similar evidence. For example, at *H4*, normal *M. trossulus* samples were identified with alleles that were exactly identical to the alleles found in both *Mytilus* BTN1 and 2. It is therefore unlikely that BTN1 and 2 arose from a single individual and multiple loci diverged to become more closely related to other normal sequences in the population. We have clarified this in the text. The addition of the fourth locus and the data from 13 *M. trossulus*-specific SNPs should also provide further evidence of these claims.

Issues that affect interpretation of the data: response should either include significant clarification or rewriting of sections:1) What is meant by "a fraction" of M. trossulus" SNPs? How many? Might this be sequencing error?

This was a brief explanation of previously published data in which a SNP array was used to assay diversity in a population of *M. edulis* individuals, and it was worded unclearly. The authors of that study noted that at some loci, there was a fluorescence signal corresponding to the *M. trossulus* allele, which should not be present in the population. More unexpected, however, was the fact that the alleles were not the expected fraction of the total alleles. A single SNP would be expected to be present at 0% (allele 1 homozygous), 50% (heterozygous with both alleles 1 and 2), or 100% (allele 2 homozygous) of the DNA at a single locus in a sample, but in these samples the fraction of the *M. trossulus* alleles were between these numbers (e.g. 30%). This suggested that the DNA sample tested could be a combination of two genomes, one a normal *M. edulis* genome and one a cancer from *M. trossulus*. This by itself is clearly not conclusive, and it led to the current study of those samples and others. We have attempted to expand and clarify our explanation of this in the text.

2) What is meant by: We sequenced…and found more than the normal two alleles in "most" of them…What is "most"? There are many such examples.

In every case of ambiguous words, such as “many” or “most,” we have replaced them with specific numbers. In the case specifically mentioned, “most” of four samples was three samples.

3) The qPCR results from attempted cloning of disease alleles is confusing. Finding that two of four disease M. chilensis animals have high levels of the cancer associated allele while the other two did not – are the authors suggesting yet another transmissible cancer lineage?

That is a likely possibility. Based on the complete absence of *Mytilus* BTN2 alleles at all four loci in those two *M. chilensis* samples, we conclude that those two samples do not have *Mytilus* BTN2. They therefore either 1) were misdiagnosed, 2) have conventional cancers that are not transmissible, or 3) have yet another transmissible cancer lineage that has not yet been described. We have clarified this in the text.

4) Given the inability to consistently amplify cancer alleles in various samples, it is unclear how any conclusions regarding the contribution of the Canadian samples and M. trossulus can be made.

We are unable to consistently clone alleles from some cancer samples, but we are able to consistently amplify cancer-associated alleles by qPCR from all samples diagnosed as cancerous from *M. edulis* and *M. chilensis*, with the exception of the two *M. chilensis* samples that are negative at all four loci (and which therefore do not have *Mytilus* BTN2). We included analysis of the Canadian *M. trossulus* samples to determine if the *Mytilus* BTN2 alleles could be detectible in the normal population or in the samples positive for *Mytilus* BTN1. We did not find any *M. trossulus* samples with alleles that match *Mytilus* BTN2 at all loci (indeed no Canadian *M. trossulus* sample amplified at all with the *EF1alpha* cancer-specific primers). This showed that these diagnostic primers consistently identify *Mytilus* BTN2, and that *Mytilus* BTN2 was not found in that *M. trossulus* population and is distinct from *Mytilus* BTN1.

5) It is also unclear how to interpret the fact that H4 cancer associated alleles are found in several normal M. trossulus animals. How many is "some?"

We have clarified ambiguous words in the text, like “some.” Specifically, since all transmissible cancers arose from a single normal individual that had the first cancer that gave rise to the lineage, it is expected that alleles nearly matching the cancer could be found in normal animals in the population. Therefore, it is expected that some normal *M. trossulus* animals would have *H4* alleles nearly identical to *Mytilus* BTN1 and some other would have alleles nearly identical to *Mytilus* BTN2. Notably, no normal animal has yet been found with *Mytilus* BTN2 or BTN1-like alleles at all loci. The two individuals in which these cancer lineages first arose would, but those two individuals are long dead. We have elaborated on this in the text.

6) Sequencing less than 1kb over 3 loci makes drawing conclusions about origin and hybridization tricky. As example, how did they determine that the EF1a 'cancer' alleles were not copy number variants in the host? Were noncancerous tissues tested for absence of these alleles in carrier individuals? Specifically, looking only at normal alleles, there is a clear grouping based on species of origin in a phylogenetic tree There are three exceptions to this clustering, from one normal M. trossulus (MW59) and two diseased M. chilensis (Mch41 and Castro26) with some distinctly M. edulis-like alleles. These may reflect introgression or true hybrid individuals-MW59 also has an EF1α allele corresponding to M. edulis and likely represents a hybrid with this species, which has been introduced into Canada (Crego-Prieto et al., 2015)." This sort of noise, although expected, becomes problematic when it encompasses a third of the data. Please explain in more detail your rationale for your conclusions and include a discussion of caveats.

As noted above, the gene trees shown in this study would be insufficient to resolve the coalescence and complete evolutionary history of the three sexual *Mytilus* species tested here.

We do have evidence of both introgression (at the *EF1alpha* locus) as well as hybrid individuals (such as MW59, noted above). However, the data on the cancer-associated alleles strongly support clonal, asexual lineages, and all four trees are completely concordant (with the exception of the finding of mitochondrial recombination in two individuals).